# *Bauhinia forficata* Link, Antioxidant, Genoprotective, and Hypoglycemic Activity in a Murine Model

**DOI:** 10.3390/plants11223052

**Published:** 2022-11-11

**Authors:** Erika Anayetzi Chávez-Bustos, Angel Morales-González, Liliana Anguiano-Robledo, Eduardo Osiris Madrigal-Santillán, Cármen Valadez-Vega, Olivia Lugo-Magaña, Jorge Alberto Mendoza-Pérez, Tomás Alejandro Fregoso-Aguilar

**Affiliations:** 1Escuela Nacional de Ciencias Biológicas, Instituto Politécnico Nacional, Department de Fisiología. Av., Wilfrido Massieu S/N, Col. Nueva Industrial Vallejo, Alcaldía Gustavo A. Madero, Ciudad de México C.P. 07700, Mexico; 2Escuela Superior de Cómputo, Instituto Politécnico Nacional, Av. Juan de Dios Bátiz S/N Esquina Miguel Othón de Mendizabal, Unidad Profesional Adolfo López Mateos, Ciudad de México C.P. 07738, Mexico; 3Escuela Superior de Medicina, Laboratorio de Farmacología Molecular, Instituto Politécnico Nacional, Alcaldía Miguel Hidalgo, Ciudad de México C.P. 11340, Mexico; 4Laboratorio de Medicina de Conservación, Escuela Superior de Medicina, Instituto Politécnico Nacional, México, Plan de San Luis y Díaz Mirón, Col. Casco de Santo Tomás, Alcaldía. Miguel Hidalgo, Ciudad de México C.P. 11340, Mexico; 5Área Académica de Medicina, Instituto de Ciencias de la Salud, Universidad Autónoma del Estado de Hidalgo, Ex-Hacienda de la Concepción, Tilcuautla, San Agustín Tlaxiaca C.P. 42080, Mexico; 6Preparatoria Número 1, Universidad Autónoma del Estado de Hidalgo, Av. Benito Juárez S/N, Constitución, Pachuca de Soto C.P. 42060, Mexico; 7Escuela Nacional de Ciencias Biológicas, Instituto Politécnico Nacional, Department de Ingeniería en Sistemas Ambientales. Av., Wilfrido Massieu S/N, Col. Nueva Industrial Vallejo, Alcaldía Gustavo A. Madero, Ciudad de México C.P. 07700, Mexico

**Keywords:** *Bahuinia forficata*, diabetes, antioxidant, hypoglycemic activity, mice

## Abstract

*Bauhinia forficata* L. is a tree used in alternative medicine as an anti-diabetic agent, with little scientific information about its pharmacological properties. The hypoglycemic, antioxidant, and genoprotective activities of a methanolic extract of *B*. *forficata* leaves and stems combined were investigated in mice treated with streptozotocin (STZ). Secondary metabolites were determined by qualitative phytochemistry. In vitro antioxidant activity was determined by the DPPH method at four concentrations of the extract. The genoprotective activity was evaluated in 3 groups of mice: control, anthracene (10 mg/kg), and anthracene + *B. forficata* (500 mg/kg) and the presence of micronuclei in peripheral blood was measured for 2 weeks. To determine the hypoglycemic activity, the crude extract was prepared in a suspension and administered (500 mg/kg, i.g.) in previously diabetic mice with STZ (120 mg/kg, i.p.), measuring blood glucose levels every week as well as the animals’ body weight for six weeks. The extract showed good antioxidant activity and caused a decrease in the number of micronuclei. The diabetic mice + *B*. *forficata* presented hypoglycemic effects in the third week of treatment, perhaps due to its secondary metabolites. Therefore, *B. forficata* is a candidate for continued use at the ethnomedical level as an adjuvant to allopathic therapy.

## 1. Introduction

According to the International Diabetes Federation, currently 537 million persons suffer from diabetes, which corresponds to 10.5% of the world’s population, and it is expected that by the year 2030, there will be 643 million, and by 2045, there will be 783 million patients with diabetes [1]. On the other hand, in Mexico in 2021, 14.1 million patients were diagnosed, ranking 7th worldwide. It is expected that by 2045, this amount will increase to 21.1 million [1]. Likewise, in the National Survey of Health and Nutrition in Mexico (ENSANUT) it was found that in 2018, the population with diabetes was 16.8%, while by 2020 it had increased to 15.7% [2,3]. For these reasons, it is clear that this chronic-degenerative disease (CDD) is a global health problem that has worsened with the arrival of the COVID-19 pandemic, as it was found that patients with COVID-19 who were hospitalized and had a diabetic comorbidity accounted for between 17 and 30% of all admissions to health centers around the world during 2020–2021 [4]. As an example of the latter, in a study conducted with patients hospitalized for COVID-19 in 2020 in Mexico, from a sample of 89, 756 patients, 17.5% also suffered from diabetes [5]. If we add to all of the latter the fact that Mexico is among the developing countries, with a high percentage of the population living in poverty, it is clearly understood that many of these low-income persons make use of alternative treatments for diabetes [6].

In México, herbalism has been employed since pre-colonial times to treat all types of diseases, including chronic degenerative diseases (CDD) such as diabetes [7,8]. In this regard, species such as nopal (*Opuntia* spp.) are widely utilized to lower blood glucose levels [9,10]. It has been found that, among the secondary metabolites contained in this plant, we find ascorbic acid, flavonoids, and phenolic acids [9] such as catechol, cinnamic acid, and 3-phenylpropionic acid, among others [11]. Another example is represented by the leaves of the Neem tree, which contain quercetin-3-O-glucoside, quercetin-3-O-rutinoside, and rutin derivatives, azadirachtin, and nimbidiol, among others [12,13]. Thus, there are many examples of plants that are used in México to treat diabetes [14], many of these originating in this country and others brought from other countries. In this regard, in some regions of Mexico, such as the state of Nuevo León and the capital Mexico City, the leaves of the plant *Bahuinia forficata* Link (Cow’s paw) are used to treat diabetes [15]. This species is thought to be an evergreen tree native to Asia or South America; it is a tree in the Fabaceae family and the tribe Cercideae, whose leaves are petiolate in the shape of a heart at their base and resemble the hoofprints left by cattle [15,16]. In various communities, therapeutic properties are attributed to it, such as diuretic, healing, antiseptic, astringent, and hypoglycemic [15]. However, although there are some publications on its different pharmacological and foraging properties [17], there are few recent studies, to our knowledge, on the therapeutic properties of this tree in Mexico. Studies have reported its anti-cancer properties [18] and its use for the treatment of infections, pain, inflammation, microbial diseases, malaria, etc., all of which are likely due to its content of secondary metabolites such as flavonoids [19]. It has also been found effective for treating cardiovascular disorders (CVD) [20] and CDD such as diabetes [21,22]. In this regard, we consider that by testing the hypoglycemic properties of this species, this could be added to the therapeutic properties mentioned above and could make this plant a multifunctional tool in the field of traditional and allopathic medicine.

Considering all of the above and the knowledge that one of the main signs of diabetes is hyperglycemia [1], this work aimed to conduct a series of experiments to determine whether *Bahuinia forficata* Link possesses antioxidant, genoprotective, and hypoglycemic properties in a murine model of chemical diabetes. For this, albino Swiss mice were chosen because of their size, which is suitable for experimental manipulation compared to other larger models (e.g., non-human primates), their short life cycle, and their adaptability to laboratory conditions.

## 2. Results

### 2.1. Qualitative Phytochemical Analysis

Thirty-one chemical reactions were carried out to identify the main secondary metabolites of the methanolic extract of the leaves of *B. forficata* Link. Table 1 summarizes the chemical reactions in which a positive result was observed for the presence of secondary metabolites in the extract.

### 2.2. Spectroscopic Analysis

Figure 1 depicts the spectrum obtained by means of a Fourier-transform infrared spectroscopy (FTIR) analysis. At the wavelength 630–735 cm^−1^, a vibration of C-H CIS bonds was located, and at the wavelength close to 1638 cm^−1^, the carbonyl functional group was located. Between 2800 and 2900 cm^−1^, C-C-H bonds were detected, and in 3295 cm^−1^, evidence of the functional carboxyl group or perhaps that of the bridged hydroxyl was found.

Several aliquots of *B. forficata* extract were taken where there was greater detection of functional groups to be subjected to other spectroscopic tests. Figure 2 presents the spectrum obtained with HPLC coupled to mass spectroscopy, where the main peak with a higher area (retention time of 3.805 min) is associated with a higher concentration of glycoside-type compounds (red circle) detected. Different reference evidence from previously published works confirms this result [23].

In several of these aliquots, when subjected to the additional HPLC-MS technique, mass peaks detected have a corresponding peak in 95% of the samples, with a flavonoid similar to kaempferol bonded with one or two glucose or arabinoside units, so the mass fragments observed of 286 *m*/*z*, 466 *m*/*z*, and 576 *m*/*z* are closely related to kaempferol derivatives such as kaempferol-3-O-arabinoside, astragalin, and kaempferitrin (Figure 3).

### 2.3. Evaluation of In-Vitro Antioxidant Activity

Figure 4 summarizes the antioxidant activity of the four concentrations of *B. forficata* tested by the in vitro DPPH method, where the percentage of inhibition of the presence of DPPH by the extract was evaluated. Ascorbic acid (2%) was employed as the reference standard, and it reached its maximal inhibition in the presence of DPPH (90.84%) when 15 min of the reaction had elapsed. The first three concentrations (50, 25, and 12.5 mg/mL) presented about 80% inhibition in the presence of DPPH in the first 5 min of the reaction and remained stable, with slight decreases at the end of the reaction (78.78%). Paradoxically, the lowest concentration (6.25 mg/mL) exhibited an inhibition in the presence of DPPH of less than 50% during the first 5 min of the reaction. Notwithstanding this, it achieved the highest percentage of inhibition (88%) within 10 min of the reaction, remaining at that level throughout the rest of the measurement time.

### 2.4. Assessment of Genoprotective Activity

Figure 5 presents the number of micronuclei in mouse peripheral blood over a period of 2 weeks, the first with the administration of anthracene and/or *B. forficata* extract (500 mg/kg), and the second without the administration of treatment. Anthracene was used because it is a mutagenic agent with a low mortality risk when used in low doses and for a short period of time. The group of mice administered anthracene (10 mg/kg) had the highest number of micronuclei on day 3 of administration, and that number decreased slightly when the mutagenic agent was no longer administered. On the other hand, animals administered with anthracene plus *B. forficata* extract had a significantly lower number of micronuclei on days 2 and 3 of the administration (19 micronuclei; *p* < 0.05; two-way repeated measures ANOVA), and when the treatment was no longer administered in week 2, that number continued to decrease significantly (two micronuclei).

### 2.5. Assessment of Hypoglycemic Activity

Because during diabetic states a loss in body weight can also occur, in this experiment, the body weight of the mice was also measured weekly during the 36-day treatment. Figure 6 shows changes in the body weight of the mice under different treatments. Diabetic mice administered with *B. forficata* prevented weight loss caused by Streptozocin administration in weeks 5 and 6 of treatment (*p* < 0.05 with a two-way repeated measures ANOVA) and, although the animals in that group had a lower weight than the controls, this was not significant (*p* > 0.05).

Figure 7 shows changes in the blood glucose values of mice under different treatments. Statistical analysis found significant differences among treatments (*p* < 0.001; two-way repeated measures ANOVA), between weeks and measurements (*p* < 0.001), as well as a significant interaction between the treatment factor and the weeks of measurement (*p* < 0.001). Overall, the administration of *B. forficata* extract to diabetic mice significantly decreased blood glucose levels from week 2 of treatment (*p* < 0.05, the post-hoc multiple comparisons test of Student–Newman–Keuls). It is noteworthy that in week 3 of treatment, the animals’ glucose levels were equal to those of the control group, and although they rose again later, this increase was significantly lower (*p* < 0.05) than that observed in the diabetic group.

## 3. Discussion

The presence of flavonoids, saponins, and tannins, among other secondary metabolites, in the methanolic extract of *Bahuinia forficata* Link leaves could aid in explaining the effects found in this work, considering that these polar compounds possess many pharmacological properties that may be in opposition to the mechanism of action of the diabetogenic agent used, streptozotocin (STZ). Several mechanisms of action for streptozotocin have been described, including DNA methylation, the activation of poly ADP-ribosylation that causes NAD^+^ and ATP depletion, the formation of free radicals, and the release of nitric oxide, all of which contribute to the death of pancreatic beta cells [24,25]. In addition, STZ has been found to inhibit the enzyme O-GlcNAcase (N-acetyl-β-D-glucosaminidase), increasing the O-glycosylation of the B-cells and their death. This in turn decreases insulin concentrations and causes hyperglycemia [26].

In this work, qualitative phytochemistry was employed as a primary tool that, although it does not establish the amount of secondary metabolites present in an extract, can provide an indication of which of those metabolites are present [27].

In this regard, the information obtained with qualitative phytochemistry was complemented with three spectroscopic analyses, including infrared with Fourier transform, GC-MS, and HPLC, finding that the most abundant secondary metabolite in the methanolic extract of *B. forficata* was a flavonoid very similar to kaempferol; however, perhaps the other metabolites detected qualitatively also contributed to the effects found in this work. Several of the secondary metabolites detected qualitatively in this work have been studied separately by different authors and share different mechanisms of action that could oppose the mechanism of action of STZ and therefore the onset and development of diabetes. For example, Ajebli et al. conducted a review of natural alkaloids and mentioned that these have a therapeutic effect on the treatment of diabetes through actions such as the blockade of protein tyrosine phosfatase 1B and the deactivation of dipeptydil peptidase-IV, increasing insulin sensitivity, modulating oxidative stress, and the inhibition of the enzyme α-glucosidase [28,29,30,31]. As for triterpenoids, it has been reported that these can be intermediates in phytosteroid synthesis and, in addition, it has been proposed that several of their biological effects, such as the inhibition of feeding, are due to the presence of hydroxyl groups [32]. Quinones (anthraquinones) have been linked to anticancer properties [33], inhibiting the activity of the enzyme P450 [34], with anti-inflammatory and antioxidant properties, among others, being the entire structure of the molecule plus the alizarin-type substituents responsible for this latter property [35]. Moreover, phenolic compounds and flavonoids have been reported to possess antioxidant properties and the ability to inhibit acetylcholinesterase and alpha-glucosidases [36], which are important features for explaining the results of this work. Prior studies had already mentioned that *B. forficata* has hypoglycemic, antioxidant, and diuretic effects, among others [19,37], and several of these studies also reported that a secondary metabolite of the flavonoid type, such as kaempferol, is abundant in the leaves of this species [38]. The latter was confirmed in this work when the HPLC analysis of our extract was performed. Several studies have found that kaempferol is also involved in the anticancer, anti-inflammatory, and anti-diabetic properties of many plants [39,40]. All of this serves as a basis for affirming that metabolites detected in the methanolic extract of *B. forficata* leaves oppose the mechanisms of action of streptozotocin, which was the agent used in our chemical model of diabetes. The four concentrations employed for the evaluation of antioxidant activity in vitro presented a high percentage in their inhibitory effect in terms of the presence of the DPPH radical during the 90 min that the reaction lasted, including the lowest concentration of the extract (6.25 mg/mL), which presented this effect after 5 min of starting the reaction. In all cases, this percentage was higher than that presented by the reference standard (ascorbic acid); thus, the extract was able to donate electrons (or donate hydrogen ions) to stabilize the last electron-deficient orbit in the DPPH, thus transforming it into its reduced form (DPPH-H). This experiment was performed in vitro, but the results can be scaled to what occurs in the chemical model of diabetes, where the STZ gave rise to the generation of free radicals such as the hydroxyl radical and peroxinitrites that, in excess, cause oxidative stress [22], which in turn causes damage to the DNA of the pancreatic beta cells of mice and therefore the death of those cells. It should be noted that in this work, the antioxidant activity of all concentrations utilized exceeded 80%, something similar to that found in another study, where the extract reached 75% of antioxidant activity with the same in vitro test [41]. It is important to note that this plant possesses great antioxidant activity that, as already mentioned, may be mediated by the presence of metabolites such as tannins, quinones, and flavonoids like kaempferol, since this property would oppose the oxidative stress that would be caused during the course of diabetes. In fact, one study found that *B. forficata* decreased hepatic oxidative stress in rats treated with bisphenol, decreasing malondialdehyde levels and increasing catalase activity [42]. For the evaluation of the genoprotective activity by the micronucleus technique in mouse peripheral blood, anthracene was used instead of streptozotocin because several studies employed this type of mutagenic agent in genotoxicity tests [43,44,45]. Anthracene is a mutagenic agent with low carcinogenic capacity [44], and its application in this work attempted to resemble the action of STZ on the DNA of pancreatic beta cells by methylating the DNA of these cells and causing their death. In the case of anthracene, alterations in the mitotic process would be caused, and consequently, the mature erythrocytes would preserve the remains of the genetic material in the form of micronuclei. In this experiment, it was found that the extract of *B. forficata* significantly decreased the number of micronuclei in the peripheral blood erythrocytes of mice compared to animals treated only with the mutagenic agent, an indication that the secondary metabolites of the extract of this species avoided the alterations of the genetic material caused by anthracene and perhaps also counteracted the methylization action of the STZ pancreatic-beta cells in the chemical model of diabetes. In other words, the extract of *B. forficata* did indeed exhibit genoprotective activity. This would agree with works in which anthracene has been used to evaluate genotoxicity [46] and with other studies in which variants of this technique have also been utilized to explore the genoprotective activity of *B. forficata*. For example, in 2013, a study was conducted in rats to determine chromosomal aberrations, and it was found that *B. forficata* did not present cytotoxic effects (mutagenicity) on the chromosomes of the bone marrow cells at a concentration of 4.65 g/L and even presented a significant antioxidant effect, all attributable to the presence of secondary metabolites such as kaempferol [47]. In another study, 7,12-dimethylbenz[a]anthracene (30 mg/kg) was utilized as a mutagenic agent, and the authors also found an elevation in the number of micronuclei in golden Syrian hamster erythrocytes [48]. It is important to note that, in the present work, the methanolic extract of *B. forficata* was employed in all experiments, and that there are studies that used the ethanolic extract and also reported antioxidant and antimutagenic effects, but it was also found that the ethereal extract does indeed possess genotoxic activity [49], leading us to consider that more and further studies still need to be conducted to clearly determine the genoprotective activity of this species. It is proposed herein that the methanolic extract had genoprotective activity of long duration in that it decreased the number of micronuclei in the erythrocytes of mice treated with anthracene both during the week of treatment as well as during the week in which the mutagenic agent was no longer administered.

In this work, two of the main signs of diabetes were reproduced: weight loss and hyperglycemia. Both in our chemical model of STZ and in diabetes, weight loss is due to the fact that when beta-pancreatic cells die, there is insufficient production of insulin to capture glucose in the different tissues and maintain homeostasis. This gives rise to the fact that the tissues use other sources of energy, such as fats and proteins, resulting in a loss of body mass and thereby the weight loss that is observed in this CDD [50,51,52]. In this respect, the *B. forficata* extract gave rise to oscillations in the body weight of diabetic mice, but such levels were maintained and were similar to those of the control group; they also had a protective effect against weight loss caused by the administration of STZ. The latter was also observed in another study, where the *Bauhinia variegata* extract was employed at doses ranging from 100–1000 mg/kg in type 1 and type 2 STZ-induced models of diabetes in rats [53]. The dose of *B. forficata* extract used in this work was 500 mg/kg, and it caused a maximal hypoglycemic effect during week 3 of treatment. Although this latter effect apparently began to fade from week 4, the glucose levels of the animals administered with the extract were always below those of the diabetic animals. This leads us to think that if a higher dose (e.g., 1000 mg/kg) had been used, the hypoglycemic effect would have lasted longer. Considering everything that has been commented on so far, we propose that the extract of *B. forficata* prevented the development of the mechanisms of action of the STZ. That is, the presence of secondary metabolites such as tannins and saponins, and especially those of the flavonoid type such as kaempferol, prevented the formation of free radicals (antioxidant activity), prevented DNA damage (genoprotective effect), and exerted a hypoglycemic and antihyperglycemic effect. In this regard, the hypoglycemic effect could also be due to the ability of the metabolites described to optimize the use and uptake of glucose by tissues other than hepatic and muscle cells, optimizing the activity of some enzymes involved in the metabolic pathways of glucose management. In this respect, there are relatively recent studies that propose mechanisms of action of *B. forficata* extract to explain the effects found in our study [22,54,55,56]. For example, a 2012 review conducted in Brazil with several species including *B. forficata* notes that tannins and flavonoids are responsible for antioxidant effects and that this justifies their use in the treatment of diabetes [57].

It is possible that the effects found with the administration of the *B. forficata* extract may be mediated by the combined mechanisms of action of the different secondary metabolites that were detected in the samples. It could be thought that, in the procedure for obtaining the crude extract, the properties of each identified metabolite could be modified; however, there are studies indicating that the processes of obtaining and drying the extract that are similar to those used in this study in *B. forficata* do not alter the properties of metabolites such as flavonoids (e.g., quercetin and kaempferol). In addition, when the authors administered that extract at a dose of 200 mg/kg iv to Wistar rats, they observed decreases of between 46.42 and 48.17% in glucose levels after the day of administration [21]. Additionally, returning again to kaempferol, there is evidence that this metabolite decreases insulin resistance through the downregulation of IκBα and the inhibition of NF-κB pathway activation [58,59]. This supports the notion that metabolites from the dry extract of *B. forficata* also exert an influence on the expression of certain metabolic pathways that may be involved in diabetes. Furthermore, as another study points out, the dry extract of the leaves of *Bauhinia holophylla* (400 mg/kg) presented hypoglycemic activity through the inhibition of glycogen synthase kinase 3-beta and an activation in glycogenesis in mice treated with STZ [60].

Taking into account all of the previously mentioned information, we support the idea of using the *B. forficata* extract in the treatment of diseases such as diabetes in countries such as México, where a high percentage of the population makes use of herbalism. In fact, in other South American countries, the commercialization of this species in naturist preparations is already being promoted, and there are already several patents for this purpose [61]. In addition, although in several of the studies the authors used it in models of type 2 diabetes, we think it could be utilized to treat diabetes in general, as indicated by another, more recent study, in which the authors used capsules of *B. forficata* extract and found that the manner of preparing and administering the extract did not affect the therapeutic properties of the secondary metabolites [62]. In addition, after it was administered every 2 days (as it was in this work) for 3 months (herein, we administered it for 1 month and a half) to 25 volunteer patients, significant decreases in triglyceride and cholesterol levels were found [63].

## 4. Materials and Methods

### 4.1. Acquisition of the Plant Species and Processing in the Laboratory

The leaves and stems of *Bahuina forficata* Link were obtained from a legally accredited business located in the state of Nuevo León, México, and were taken to the Hormones and Behavior Laboratory of the Department of Physiology of the National School of Biological Sciences to complete their environmental drying for 1 week. The material (both leaves and stems) was then crushed and macerated in methanol for 1 week. The macerate was subjected to reduced pressure distillation (Rotavaporator Prendo^©^ Model 1750, Prendo, Puebla, México) to remove the methanol; the distillate was left to air dry for 1 week to obtain the dry crude extract of leaves and stems combined.

Solubility tests were performed to select the solvent to make a stock suspension of this crude extract at 50 mg/mL, and it was found that the best solvent to resuspend the crude extract for intragastric administration in mice was distilled water.

### 4.2. Chemicals Used in the Experiments

The chemicals 2,2-diphenyl-1-picrylhydrazyl (DPPH) and streptozocin (STZ) were purchased from Sigma-Aldrich. Giemsa dye was purchased from Hycel (México). Methanol was purchased from Golden Bell Co. (Mexico City, México).

### 4.3. Animals in Laboratory Settings

In this study, male Swiss albino mice (weighing 25–30 g each) of the NIH strain were used. They were obtained from the official supplier of the National School of Biological Sciences (ENCB) of the National Polytechnic Institute and housed in the animal chamber of the Department of Physiology of the ENCB, Zacatenco campus for acclimatization in communal acrylic cages (48 cm long × 22 cm wide × 20 cm high), with water and food *ad libitum* and a light-dark cycle of 12:00 (lights turning on at 08:00), as well as a room temperature of 22 ± 2 °C and under standard humidity conditions. The animals were handled and subjected to experimentation according to Mexican standards (NOM-033-ZOO-1995, NOM-062-ZOO-1999, NOM-087-ECOL-1995) and the international bioethics standards currently in force.

### 4.4. Qualitative Phytochemical Analysis

Samples of the crude extract of the leaves and stems of *B. forficata* were taken and subjected to the various qualitative chemical reactions described in other works [64,65] to determine secondary metabolites, mainly of a polar nature, which were identified by changes in coloration, the formation of precipitates, foaming, etc.

### 4.5. Spectroscopic Analysis

Fractions of the dry extract of *B. forficata* were taken and stored in vials for a series of general spectroscopic analyses. The samples were analyzed at the Center for Nanosciences and Micro and Nanotechnologies of the National Polytechnic Institute (Mexico City, México). For the analysis of high-performance liquid chromatography (HPLC), an Agilent model 1260 Infinity II device with a variable UV-Vis wavelength detector and a wavelength range of 190 to 600 nm was employed. Also, it is coupled to an Agilent mass spectrometer. Parameter conditions used were: Mobile phase, 50% Water, 50% MeOH, and 1% FA; a flow rate of 0.4–0.5 mL/min; a stop time of 25 min; a needle wash mode and standard wash injection volume of 5 µL; a column temperature of 30 °C; UV detection at 280 nm/4 nm; Ref.: OFF > 0.025 min (0.5 s response time) (10 Hz); an MS detection acquisition mode at MS1; a minimum range of 100 *m*/*z* and a maximum range of 7000 *m*/*z*; an ion polarity at positive source parameters; a gas temperature of 290 °C; a gas flow rate of 14 L/min; a nebulizer set at 20 psig; a sheath gas temperature of 400 °C; a sheath gas flow rate of 12 L/min; scan source parameters at a Vcap of 5000, V Nozzle voltage of 2000 V, fragmentor of 500 V, skimmer of 10, octopole RF peak of 750; and reference masses of 922.0098 and 1821.9523 T.

Thirty µL of sample was injected into an Agilent ZORBAX Rx-SIL, 100 × 3.0 mm, 1.8 µm column, with a flow rate of 0.4 mL/min and a mixture of 50% methanol and 50% water. A scanning spectrum with an UV-Vis Agilent 8453 spectrophotometer was obtained for the crude extracts in order to establish the absorption wavelength to be used for the HPLC variable wavelength detector, which correspond with the higher absorption peaks observed, and also to give brief information about the concentration of the compounds mixed in the crude extract. Overall, the determined concentrations of analytes ranged from 0.0530 mg/mL–0.0002 mg/mL (before dilution). In addition to the lengths of 270 and 560 nm, a wavelength of 214 nm was also utilized to verify the presence of hypsochromic shift changes due to compounds with chromophore effects.

For the Fourier transform infrared analysis (FTIR) technique, an Espertoi Lambda Series 5000 device was used.

### 4.6. Evaluation of In Vitro Antioxidant Activity

Four concentrations of the crude extract of *B. forficata* (50, 25, 21.5, and 6.25 mg/mL) were prepared in methanol to determine its antioxidant activity using the in vitro method of DPPH (2,2-diphenyl-1-picrylhydrazyl). A solution of 0.01 g of DPPH in 25 mL of methanol was prepared.

This method was performed on a UV-VIS spectrophotometer (Velab^©^; VE-5100UV, Velaquin, México city, México); quartz cells containing 1850 μL of methanol, 140 μL of DPPH free radical, and 10 μL of some of the concentrations of the crude extract diluted in methanol were prepared, and absorbance was measured at 517 nm of each solution. The absorbance value was recorded at 0 (without extract sample) and at 1, 5, 15, 20, 30, 60, and 90 min. These readings were taken in duplicate for each time, and the average value of each absorbance was substituted in Equation (1) to calculate the % of inhibition in the presence of DPPH.
% DPPH inhibitión = (Absm_t = 0_ − Absm_t = n_/Absm_t = 0_) × 100(1)
where: Absm_t = 0_ = sample absorbance at zero time (without extract). Absm_t = n_ = sample absorbance at “n” time (with extract).

The percentage data were plotted for each concentration of the extract and compared against the % of inhibition in the presence of DPPH obtained for a reference solution (2% ascorbic acid).

### 4.7. Assessment of Genoprotective Activity

Three groups of male Swiss albino mice were formed and housed in three cages (35 cm long × 25 cm wide × 12 cm high), each containing six mice. Each group received one of the following treatments: (i) Control; administration of vehicle (mineral oil intragastrically (i.g.)) every 2 days for 1 week; taking a blood smear every 2 days for 2 weeks; (ii) Anthracene (10 mg/kg, i.g.), administration of anthracene dissolved in mineral oil, every 2 days for 1 week and with a blood smear, every 2 days for 2 weeks, and (iii) Anthracene + *B. forficata* (500 mg/kg, i.g.), administration of anthracene plus *B. forficata* extract every 2 days for 1 week and with a blood smear every 2 days for 2 weeks. All blood smears were fixed in methanol (6 min) and colored with Giemsa (40 min) to be evaluated using the micronucleus detection technique in mouse peripheral blood [43] and to be analyzed under the optical microscope (VELAB^©^; VE-M5, McAllen, TX, USA) with 100× magnification (immersion).

### 4.8. Assessment of Hypoglycemic Activity

Three groups of male Swiss albino mice were formed and housed in three cages (35 cm long × 25 cm wide × 12 cm high), each containing six mice. Each group received one of the following treatments: (i) Control mice were administered i.g. with the plant-extract dissolution vehicle every 2 days (saline, 0.9%; vol = weight/1000); each week, blood glucose was measured with a commercial device (Optium FreeStyle^©^, Roche, Boston, MA, USA), making a small cut in the distal part of the mouse’s tail to drain one drop of blood. Additionally, the body weight of the mice was recorded (prior fasting, <12 h) for 36 days; (ii) Diabetic mice were administered with a single intraperitoneal dose (i.p.) of streptozotocin (STZ; 120 mg/kg, dissolved in citrate buffer). After 1 week of administration, blood glucose was measured with a commercial device (Optium FreeStyle^©^). A mouse was considered diabetic (hyperglycemic) when its blood glucose levels reached 150 mg/dL or higher. Additionally, the body weight of the mice was recorded. These animals were administered i.g. every 2 days with a vehicle and with a weekly measurement of glucose and triglycerides (prior fasting, <12 h) for 36 days; (iii) Diabetic + *B. forficata* extract (500 mg/kg, i.g.), mice administered i.g. every 2 days with the methanolic extract of *Bahuina forficata* (500 mg/kg) and with a weekly glucose measurement (prior fasting, <12 h) for 36 days.

### 4.9. Statistical Analysis

The data obtained in the experiments of genoprotective activity and hypoglycemic effect were analyzed with SigmaStat ver. 12.0 statistical software, utilizing repeated measures two-way ANOVA and the Student–Newman–Keuls post-hoc test to determine significant differences among the different groups. In all cases, a level of α = 0.05 was employed as the criterion for establishing statistically significant differences.

## 5. Conclusions

In this work, evidence has been provided of the antioxidant, genoprotective, and hypoglycemic activity of the methanolic extract of *B. forficata* leaves and stems. We think that it could be used for both type 1 and type 2 diabetes. However, more in-depth studies on these properties still need to be conducted, and one of these would be to isolate the secondary metabolites found in the dry extract in order to administer it in the chemical model with STZ to determine whether the therapeutic effect is determined by any of the metabolites detected with the phytochemistry or whether it is due to the set of these present in the dry extract.

In summary, this study proposes the use of *B. forficata* extract as an alternative co-adjunctive therapy for the treatment of diabetes, without leaving allopathic therapy to one side, which could aid in improving the quality of life of patients with this pathology and perhaps reduce the treatment cost and the severity of the side effects of several of the drugs used.

## Figures and Tables

**Figure 1 plants-11-03052-f001:**
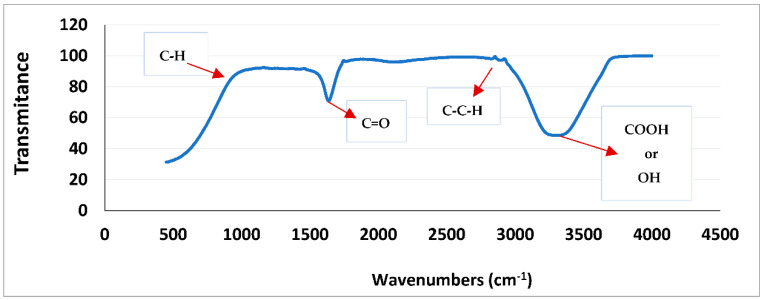
**The** spectrum obtained through Fourier-transform infrared (FTIR) analysis of the methanolic extract of *Bahuinia forficata* leaves and stems.

**Figure 2 plants-11-03052-f002:**
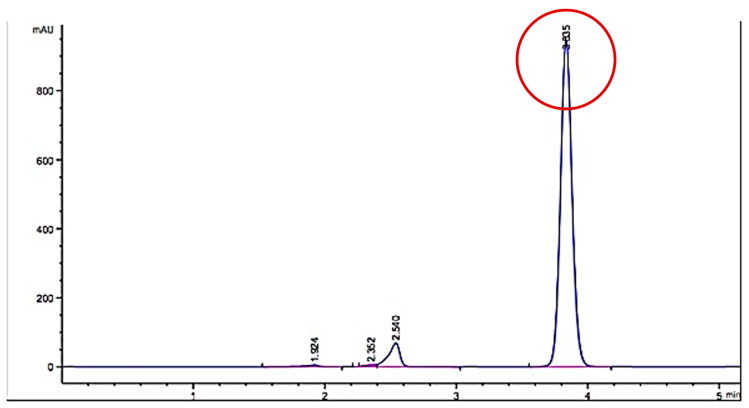
Spectrum obtained with coupled HPLC-mass spectroscopy from some aliquots of the *Bahuinia forficata* extract. Red circle denotes the retention time of 3.805 min corresponding to glycoside-type compounds.

**Figure 3 plants-11-03052-f003:**
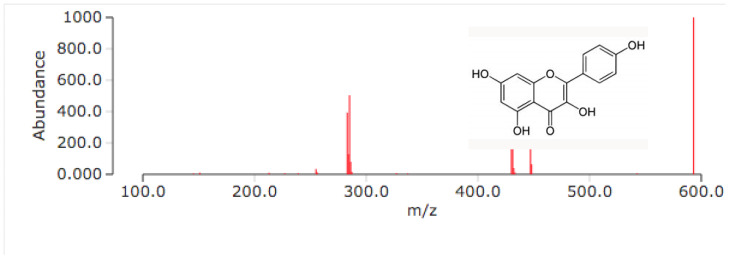
The spectrum obtained with HPLC showing the zone of the functional groups that may correspond to secondary metabolites derivatives of kaempferol.

**Figure 4 plants-11-03052-f004:**
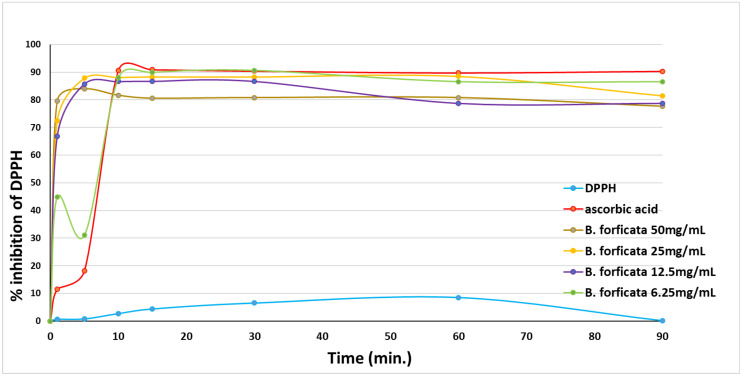
Percentage of inhibition in the presence of DPPH of four concentrations of *B. forficata* during 90 min of the reaction. Each point represents the average of two absorbance measurements (517 nm) substituted in Equation (1), as described in the methodology.

**Figure 5 plants-11-03052-f005:**
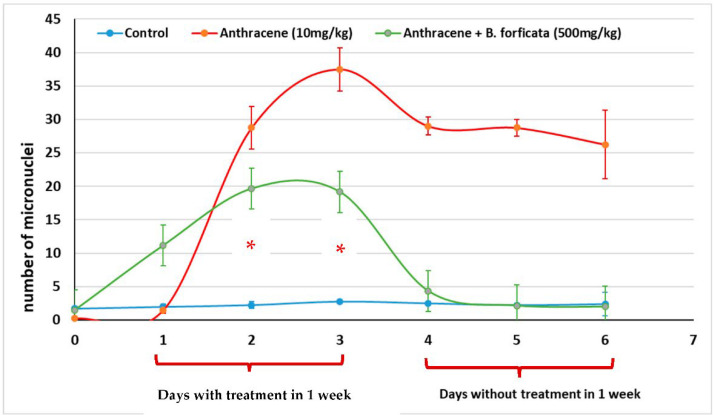
Genoprotective activity measured as the number of micronuclei in peripheral mouse blood. Data is expressed as mean ± standard error of the mean (SEM). * denotes a *p* < 0.05; comparison of anthracene + *B. forficata* vs. anthracene as determined by a two-way repeated measures ANOVA.

**Figure 6 plants-11-03052-f006:**
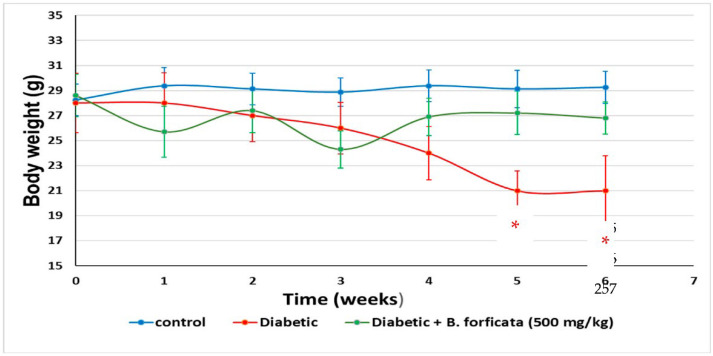
Changes in the body weight of mice under different treatments for 36 days (6 weeks). Data is expressed as mean ± SEM. * denotes a *p* < 0.05; comparison of diabetic + *B. forficata* vs. diabetic and control groups as determined by a two-way repeated measures ANOVA.

**Figure 7 plants-11-03052-f007:**
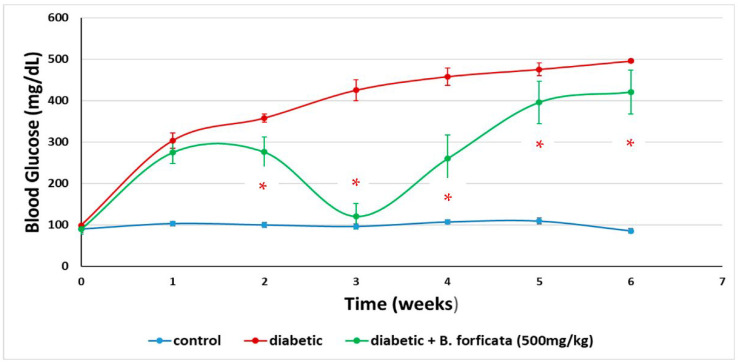
Glucose levels recorded in mice under different treatments for 36 days (6 weeks). Data is expressed as mean ± SEM. * denotes a *p* < 0.05; comparison of diabetic + *B. forficata* vs. diabetic as determined by a two-way repeated measures ANOVA.

**Table 1 plants-11-03052-t001:** Secondary metabolites detected by qualitative phytochemistry in the dry extract of the combination of leaves and stems of *Bahuinia forficata*.

Secondary Metabolite	Test
Alkaloids	Dragendorff
	Sonnenschain
	Wagner
Flavonoids	Shinoda (Flavones)
	10% sodium hydroxide (Flavonols)
Saponins	Liebermann Buchard (triterpenoids)
	Rosenthaler (triterpenoids)
Quinones	Ammonium hydroxide (Anthraquinones)
	Börntraguer (Anthraquinones)
Reducing sugars	Fehling
	Benedict
Tannins	1% ferric chloride (Phenolic compounds)

## Data Availability

Not applicable.

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
