# Peer review of "Bauhinia forficata Link, Antioxidant, Genoprotective, and Hypoglycemic Activity in a Murine Model"

_plants, 2022, doi:10.3390/plants11223052_

Round 1
Reviewer 1 Report
Authors present their studies on Asia and South America originated tree Bauhinia forficata Link. Especially they focus on antioxidant, genoprotective and hypoglycemic activity in a murine model, using a methanolic extract. The work is clearly presented in good English and could be of interest for the Pharmaceutical and Food Industry. But several mistakes must be eliminated and some facts explained more precise before publication.
Formal
1.) Please shorten ABSTRACT to max. 200 words (see guide for authors).
2.) Line spacing shoul be unified. It changes several times within the document (see lines 51 - 67; 90 -93; 299 - 315 ; 398 ff).
3.) Is figure 1 the original IR-spectrum? I never saw a print-out with such a fat recorded line. If not: please exchange for the original IR-spectrum.
4) Figures 2 and 3 are swapped. They need to be exchanged.
5.) Some additional slight errors in the text and several hints/proposals are indicated directly in the draft.
Content
1.) You mention Murine model in the heading. It does not appear in the text. You should shortly explain /define it and what is the advantage of your selected model.
2.) You should also describe in DISCUSSION shortly, what genoprotective activity means here.
3.) Lines 84 / 89 ff: In my opinion there is a lot of significant studies in this area, rather near to the proposed article of you (see ref 21 and 22 and literature cited). All mentioned properies (antioxidant, genoprotective, and hypoglycemic) are tested elsewhere by other groups. Please point out what is new and complementary in your work.
4.) Line 143: Which samples? Please describe more detailed.
5.) Figure 5: Are the experiments summarized in figure 5 repeatable? How often have they been performed?
6.) Line 216 ff: You should briefly explain the use of anthracene and the meaning and origin of the micronuclei here. It is explained in the discussion. Maybe you transfer it from there (lines 369 ff) to here.

Reviewer 2 Report
Abstract
1. Extract fractions were used to determine secondary metabolites by qualitative phytochemistry. five concentrations of the crude extract of leaves and stems were prepared (50, 25, 12.5, 6.25 y 3.125 mg/mL) for the evaluation of in vitro antioxidant activity by the DPPH method.
What is "y". and five concentrations..... should be Five concentrations.
2. Control (vehicle), Anthracene (10 mg/kg), and Anthracene (10 mg/kg) + B. forficata extract (500 mg/kg) and the presence...
Which extract leaves or stems??????
3. To determine the hypoglycemic activity, the crude extract was prepared in a suspension and administered...
Which extract leaves or stems??????
4. The overall abstract is not clear and should be re-written with more results describtion
Results
5. Table 1. should be inserted as supporting data not in the manuscript itself.
6. Flavones, flavonoids, triterpenoids (saponins), antrhaquinones, and phenolic compounds are examples of secondary metabolites of a polar nature with antioxidant and hypoglycemic properties.
Not inportant in this position
7. UV for total extract of what??? leaves or stems????
8. UV for crude extract???? what the benefits of it.
9. GMS is not suitable at all for polar compounds. it is suitable only for volatile constituents. I think it should be deleted, HPLC is enough
10. In abstract "five concentrations of the crude extract of leaves and stems were prepared" and in results, the authors described one extract?????? which one
11. The authors studied one extract or two??????
Materials
12. The leaves and stems of Bahuina forficata Link were obtained from....
You used one or two plant parts????
13. This sentence described that the authors use two plant parts. Are these two parts were extracted to two extracts or what????
14. The extraction process should be re-written?
Round 2
Reviewer 2 Report
I think the manuscript can be accepted in the present format